# Manganese Exposure and Metabolic Syndrome: A Systematic Review and Meta-Analysis

**DOI:** 10.3390/nu14040825

**Published:** 2022-02-16

**Authors:** Martin Ming Him Wong, Kwan Yi Chan, Kenneth Lo

**Affiliations:** 1School of Professional and Continuing Education, The University of Hong Kong, Hong Kong, China; martinwong1112@gmail.com; 2School of Public Health and Primary Care, The Chinese University of Hong Kong, Shatin, Hong Kong, China; apple_chan11@hotmail.com; 3Department of Applied Biology and Chemical Technology, The Hong Kong Polytechnic University, 11 Yuk Choi Road, Hung Hom, Kowloon, Hong Kong, China; 4Research Institute for Smart Ageing, The Hong Kong Polytechnic University, Hong Kong, China

**Keywords:** manganese, micronutrient, metal exposure, metabolic syndrome, meta-analysis

## Abstract

Manganese (Mn) is an essential element acting as a co-factor of superoxide dismutase, and it is potentially beneficial for cardiometabolic health by reducing oxidative stress. Although some studies have examined the relationship between Mn and metabolic syndrome (MetS), no systematic review and meta-analysis has been presented to summarize the evidence. Therefore, the present review examined the association between dietary and environmental Mn exposure, and MetS risk. A total of nine cross-sectional studies and three case-control studies were included, which assessed Mn from diet, serum, urine, and whole blood. The association of the highest Mn level from diet (three studies, odds ratio (OR): 0.83, 95% confidence interval (C.I.) = 0.57, 1.21), serum (two studies, OR: 0.87, 95% C.I. = 0.66, 1.14), urine (two studies, OR: 0.84, 95% C.I. = 0.59, 1.19), and whole blood (two studies, OR: 0.92, 95% C.I. = 0.53, 1.60) were insignificant, but some included studies have suggested a non-linear relationship of urinary and blood Mn with MetS, and higher dietary Mn may associate with a lower MetS risk in some of the included studies. While more evidence from prospective cohorts is needed, future studies should use novel statistical approaches to evaluate relative contribution of Mn on MetS risk along with other inter-related exposures.

## 1. Introduction

Metabolic syndrome (MetS) is defined as a cluster of metabolic alterations that contributes to a higher risk of cardiovascular disease (CVD), type 2 diabetes (T2D), and all-cause mortality [1,2]. According to data from the 2011–2016 National Health and Nutrition Examination Survey (NHANES), more than a third of adults in the United States (U.S.) have MetS, and the rate can be as high as 48.6% among those aged at least 60 years [3].

In recent years, an increasing number of researchers have been investigating how manganese (Mn) is potentially beneficial for cardiometabolic health. Mn is an essential element that acts as a co-factor of superoxide dismutase, an enzyme responsible for the degradation of reactive oxygen species (ROS) [4]. Evidence from in vitro and animal studies has demonstrated that Mn supplementation could downregulate ROS generation [5], prevent endothelial dysfunction [6], and reduce the levels of serum inflammatory biomarkers [7]. However, excessive exposure to Mn from polluted air and water may lead to impaired cognitive development and Parkinson’s disease (PD), especially among workers and general populations residing near factories [4,8]. Although the mechanisms linking Mn overexposure and PD are still under investigation, a combination of mitochondrial dysfunction and oxidative stress, protein misfolding and trafficking, and neuroinflammation, may play major roles in Mn neurotoxicity [9]. Mn can be obtained from water, nuts, grains, fruits, green vegetables, and caffeinated drinks [10]. Mn exposure can also be reflected from its level from blood and urine. The half-life of blood Mn is 10 to 42 days, but that for urinary Mn is less than 30 h, indicating a more recent exposure than blood Mn [11].

Despite the increasing attention to the relationship between Mn, from various sources, and cardiometabolic health, the research evidence has not been summarized and quantified in a systematic manner. To address the knowledge gap as mentioned above, we have conducted a systematic review and meta-analysis to reveal the association between Mn exposure from diet and environment, and the risk of MetS.

## 2. Materials and Methods

### 2.1. Data Sources and Searches

As already registered in PROSPERO (CRD42021289335), we followed a standardized protocol to conduct this systematic review and meta-analysis on 5 December 2021, in accordance with the Preferred Reporting Items for Systematic Reviews and Meta-Analyses (PRISMA) statement [12] and Cochrane Handbook [13]. Two reviewers (Wong M.M.H. and Lo K.) independently conducted literature search of Embase 1910 to Present, Ovid Emcare 1995 to 2021 Week 48, and Ovid MEDLINE(R) and Epub Ahead of Print, In-Process, In-Data-Review & Other Non-Indexed Citations 1946 to 3 December 2021. Detailed keywords for the search are outlined in the Supporting Information. Search terms included a combination of synonyms of manganese (Mn) and “metabolic syndrome” [14,15,16] (Metabolic Syndrome OR MetS OR Syndrome X OR Insulin Resistance Syndrome OR Dysmetabolic Syndrome OR Reaven Syndrome OR Metabolic Cardiovascular Syndrome OR Cardiometabolic Syndrome) as adapted from relevant review articles [14,15,16,17]. All articles with English abstracts were assessed.

### 2.2. Study Selection

Studies were included if they (1) were observational studies with a cross-sectional, prospective, or retrospective design; (2) examined manganese exposure from diet or environment; (3) defined metabolic syndrome with criteria or guidelines; (4) examined the association between manganese exposure and metabolic syndrome; and (5) enrolled human participants. We excluded studies if they (1) were literature reviews, clinical trials, editorials, or abstracts from conference proceedings; or (2) did not have an abstract or full text in English.

### 2.3. Data Extraction and Quality Assessment

Data extraction was conducted independently by two investigators (Wong M.M.H. and Lo K.), and discrepancies were resolved through consensus. The following information was extracted from all eligible studies: the country where the study was conducted, study design, age and number of participants, gender ratio, and references used to define metabolic syndrome, by using a standard data extraction form created in Microsoft Excel. The exposures and outcomes of the present review included Mn exposure and MetS status. The effect measures, such as odds ratio (OR) or relative risk (RR), their 95% confidence interval (95% C.I.) and standard errors (SE) were extracted from the articles. Results without adequate information for conducting meta-analysis were described in a narrative fashion.

The methodological quality of the included studies was assessed by the Newcastle–Ottawa Scale (NOS) for observational studies, which is recommended by the Cochrane Collaboration [18]. The checklist assessed the possibility of bias in the selection, comparability, and outcomes of each study. The ratings of each item ranged from 0 to 2, with a total of 10 (9 for case-control studies). The total score for each included study was computed.

### 2.4. Data Synthesis and Analysis

The fully adjusted effect estimates for the highest versus the lowest category of exposure (Mn exposure from diet, urine, serum, or whole blood) and their associations with MetS were extracted from each included study. Random effects models using the inversed variance approach were used to pool the estimates from individual studies because of the varying population and criteria used to define outcomes. The results were summarized using forest plots. I^2^ was used to assess the heterogeneity, with an I^2^ between 50% and 90% possibly representing substantial heterogeneity [13]. To demonstrate the consistency of findings across studies, we also omitted one analysis at a time to observe how the magnitude of association and heterogeneity varied. If an included study had sex-stratified analysis, we extracted the effect estimates from each sex for meta-analysis. Meta-analyses and forest plots were performed by Review Manager 5.2.

## 3. Results

### 3.1. Characteristics of Included Studies

Figure 1 shows the selection process for the studies included in the review. A total of 49,414 participants from 12 studies (6 conducted in China, 3 in U.S., 2 in Korea, and 1 in Iran) were included. Nine of the included studies were cross-sectional, three were case-control studies, and all were published between 2013 and 2021. One study was prospective cohort in study design, but the authors only analyzed the association between Mn and MetS using baseline data, and therefore the paper was regarded as cross-sectional study [19]. NCEP ATP III was the most popular diagnostic criteria for MetS (six studies), followed by AHA/NHLBI Scientific Statement (three studies). The methodological quality of the studies ranged from 6 to 8 out of 9 (out of 10 for cross-sectional studies). The characteristics and outcome definitions of the included studies are described in the Table 1.

### 3.2. Dietary Mn and MetS

Three of the included studies examined the association between dietary Mn and the presence of MetS [21,23,29]. Among the 2111 adults that participated in the Chinese National Nutrition and Health Survey 2010–2012 [29], men with the highest Mn intake (>6.87 mg/day) had a significant lower likelihood for MetS (OR: 0.62, 95% C.I. = 0.42, 0.92), but a positive association (OR: 1.56, 95% C.I. = 1.02, 2.45) was found for women with the highest Mn intake (>5.79 mg/day). There was significant interaction between sex and dietary Mn in affecting the likelihood for MetS. On the other hand, for a case-control study conducted among 550 adults [23], the highest quartile of Mn intake was associated with a lower likelihood of MetS (OR: 0.47, 95% C.I. = 0.29, 0.79), but the result was not stratified by sex. Choi et al. analyzed the data of 5136 adults from the general population of Korea [21], but did not found a significant relationship between the highest quartile of Mn intake and MetS among men (OR: 1.01, 95% C.I. = 0.68, 1.49) nor women (OR: 0.82, 95% C.I. = 0.55, 1.22). When pooling the results from included studies (Figure 2), the overall association between the highest level of dietary Mn and MetS was not significant (OR: 0.83, 95% C.I. = 0.57, 1.21, I^2^ = 74%), and the heterogeneity across study was substantial (I^2^ > 50%). When omitting one analysis at a time, the heterogeneity remained substantial (53% to 81%). The overall association changed to significant (pooled OR: 0.72, 95% C.I. = 0.53, 0.98) only after omitting the analysis on dietary Mn and MetS among participating women in the study of Zhou and colleagues (OR: 1.56, 95% C.I. = 1.02, 2.45) [29].

### 3.3. Serum Mn and MetS

Two included studies conducted in China examined the association between Mn from serum and the likelihood of MetS [19,28]. Feng et al. analyzed the association of 11 serum metals with MetS among 4303 men who participated in the Fang Chenggang Area Male Health and Examination Survey cohort [19]. As one of the selected serum metals, Mn was only inversely associated with MetS in the second tertile (OR: 0.65, 95% C.I. = 0.43, 0.98), not the highest tertile (OR: 1.06, 95% C.I. = 0.95, 1.18) [19]. Similarly, the case-control study of 4134 adults conducted by Zhang et al. analyzed 15 serum metals and their association with MetS by putting them in the same logistic regression model [28]. Serum Mn was associated with a lower likelihood of MetS (OR: 0.78, 95% C.I.: 0.63, 0.97) only at the highest quartile (>1.69 µg/L) [28]. When pooling the results from included studies (Figure 2), the overall association between serum Mn and MetS was not significant (OR: 0.87, 95% C.I. = 0.66, 1.14, I^2^ = 47%), and the heterogeneity across studies was not substantial (I^2^ < 50%). Given the limited number of studies, we did not omit one analysis at a time for this part.

### 3.4. Urinary Mn and MetS

Four included studies examined the association between urinary Mn and the presence of MetS [22,24,25,27], and two of them provided adequate data for meta-analysis (Figure 2) [24,25], but the overall association between the highest level of Mn from urine and MetS was not significant (OR: 0.84, 95% C.I. = 0.59, 1.19, I^2^ = 0%), and the heterogeneity across studies was not substantial (I^2^ < 50%). When omitting one analysis at the time, the I^2^ value remained as 0% and the overall association remained insignificant. However, nonlinear relationship between urinary Mn and MetS was observed from some of the included studies. By analyzing the data from U.S. NHANES 2011–2016, Lo et al. observed that urinary Mn at the third quartile associated with a lower odd of MetS among overall participants (OR = 0.55, 95% C.I. = 0.32, 0.97) and men (OR = 0.40, 95% C.I. = 0.16, 0.99) [24]. From the restricted cubic spline analysis, the U-shaped dose-response relationship between urinary Mn and MetS was observed among all participants [24]. As demonstrated by posterior inclusion probabilities (PIP), urinary Mn played a less important role in development of MetS (PIP = 0.49 for Mn versus 0.54 to 0.91 for other metals) [24]. For two case-control studies (conducted in Iran and China, respectively) that were not included in the meta-analysis, per unit increment of urinary Mn did not associate with MetS [22,27].

### 3.5. Whole Blood Mn and MetS

Four of the included studies examined the association between Mn from whole blood and the presence of MetS [20,24,26,30]. For the three studies that analyzed data from U.S. NHANES, different approaches were adapted in each included study. Bulka et al. put each metal exposure separately into the logistic regression model [20], and found that blood Mn did not associate with MetS across the quartiles (OR at quartile 4: 1.04, 95% C.I. = 0.88, 1.24). Moreover, Lo et al. observed a null association between blood Mn and MetS for both male and female participants in the multi-metal model (including all whole blood metals into the regression model) [24]. They have further evaluated the relative importance of blood Mn and other metals in the association with MetS by the Bayesian kernel machine regression (BKMR) model. As measured by PIP, they found that blood Mn has the least relative importance in the presence of MetS compared with other blood metals (cadmium, mercury, lead, and selenium) [24]. Zhou et al. demonstrated that the association between blood Mn and MetS per log increment was not significant (OR: 1.22, 95% C.I. = 0.96, 1.56), which was consistent across age groups and sex [30]. In addition, they demonstrated a M-shaped association between blood Mn and MetS using restricted cubic spline analysis [30]. Rhee et al. analyzed data from the Korea NHANES 2008, and they did not find a significant association between blood Mn and MetS across the quartiles (OR at the highest quartile: 1.22, 95% C.I. = 0.76, 1.97) [26]. To perform meta-analysis, the data from Lo et al. were selected out of three studies that analyzed U.S. NHANES, because it covered a larger data set than Bulka et al., while Zhou et al. did not provide the effect estimate across quartiles. When pooling the results from the included studies (Figure 2), the overall association between whole blood Mn and MetS was not significant (OR: 0.92, 95% C.I. = 0.53, 1.60, I^2^ = 51%), and the heterogeneity across the studies was substantial (I^2^ > 50%). When omitting one analysis at the time, the overall association remained insignificant, but the heterogeneity was not substantial after excluding the analysis from the male participants of Lo’s study (I^2^ = 0%) [24], and the data from Rhee et al. (I^2^ = 6%) [26].

## 4. Discussion

In the present review, we have summarized how dietary Mn intake, and Mn levels as reflected from serum, urine, and whole blood, may associate with the risk of MetS. However, we did not find significant association between the highest level of Mn from diet, blood, nor urine with MetS from meta-analysis. The insignificant results could be explained by two reasons. First, all the included studies were cross-sectional or case control in nature, which could be biased by reverse causality, attenuating the relationship between Mn and MetS as suggested by the physiological mechanism.

Moreover, the Mn–MetS association could be non-linear, as suggested by some included studies examining Mn exposure from serum [19], urine [2,4], and whole blood [30]. From a physiological perspective, deficient and excessive Mn exposure may relate to a higher risk for metabolic syndrome. As a co-factor of Mn superoxide dismutase, Mn deficiency may increase oxidative stress by producing more ROS [8], leading to inflammation and endothelial dysfunction [31,32], and accelerate the proliferation of vascular cells and increase vasoconstriction [33,34]. On the other side, Mn shares the calcium uniporter mechanism, and the accumulation of excessive Mn may inhibit the efflux of calcium, then inhibit the respiratory chain and adenosine triphosphate production [9]. This will in turn disrupt normal mitochondrial function and also increase oxidative stress, elevating the risk of metabolic diseases [8]. With regard to the harm–benefit duality of Mn, it is necessary to identify the optimal level of exposure. Although the statistical approaches for dose-response meta-analyses have been well-established [35], as revealed by the present review, very few included studies have provided adequate information for analysis [21,28], such as the levels of Mn exposure and cases by quantiles. A more unified reporting format of results, providing adequate data for dose-response meta-analysis, will facilitate the summary of evidence on the association of Mn with MetS in the future.

Moreover, one included study investigated the effects of metal mixtures on MetS, which used the BKMR model to quantify the relative importance of each urinary/blood metal on the presence of MetS [24]. In the study conducted by Lo and colleagues, Mn from urine and blood might have less contribution to MetS risk than heavy metals such as cadmium and mercury [24]. For other metals included in this study, arsenic, cadmium, and mercury may have dose-response toxicity to adverse cardiometabolic health [36,37,38]. Meanwhile, Mn serves as both essential metal and neurotoxin depending on dosage [8], and therefore the variation in the shape of relationship may weaken the association of Mn with MetS. The rationale of using a machine-learning approach is that most previous studies have estimated the association of single-metal exposure with disease risk by adjusting for other multiple metals in the traditional regression models simultaneously, or introducing the cross-product terms [39,40]. These traditional approaches have methodological limitations and cannot address the overall effects of metal mixtures, single-metal effects, and their interactions in the high-dimensional set of correlated exposures [41]. In other words, future studies are suggested to incorporate novel statistical approaches that can account for the interactions of multiple metal exposures.

For the relationship between dietary Mn and MetS, two out of the three included studies observed significant inverse associations [23,29]. The overall association was significant after omitting the positive association between dietary Mn and MetS in women [29]. The potential inverse association between dietary Mn and MetS was consistent with the findings from several prospective cohorts on the relationship between Mn from diet and type 2 diabetes, which were conducted among general population in China and Japan [42,43], as well as postmenopausal women from the U.S. [10]. However, the results should be interpreted with caution, because only 3% to 5% of dietary Mn is absorbed from the gastrointestinal tract [9]. The observed association could be confounded by other nutrients (e.g., magnesium) from the common food sources (nuts, grains, fruits, green vegetables, and caffeinated drinks) [44]. Similar to the research on environmental exposure of Mn, future studies may adapt statistical approaches that quantify the relative importance of inter-related dietary exposure and MetS risk. One example is a recent publication that explored the association between 12 dietary factors and 10 year predicted risk of atherosclerotic cardiovascular disease using BKMR, and found that fruit intake was the strongest protective factor among men and unprocessed red meat was the most important predictor among women [45]. In addition, more evidence from prospective cohorts is needed to verify the potential effect modification by sex in the relationship between dietary Mn and MetS.

The major strength of the present systematic review and meta-analysis lies in quantifying the influence of Mn exposure from various sources, including diet, serum, whole blood, and urine, respectively, on the presence of MetS, which is the first meta-analysis conducted to summarize this relationship. It also covers various populations in the world. Nonetheless, several limitations should be noted. Firstly, the present study included only articles written in English, wherein eligible studies published in other languages might have been overlooked. However, it is arguable that many of the included studies were performed in China; therefore, the language restriction is probably not a major flaw in the present literature search. Thirdly, in the present review, all included studies were cross-sectional or case-control in nature, which are subject to reverse causality, and prospective cohort studies should be conducted to elucidate the temporal relationships. Lastly, different definitions of MetS should also be adapted in the same study to identify how the associations may differ.

## 5. Conclusions

The overall association between dietary, serum, urinary, and whole blood Mn and MetS was not significant, which might be attributed to the inconsistency in epidemiological findings. However, urinary and blood Mn may have a non-linear relationship with MetS, and higher dietary Mn may associate with a lower risk of MetS in some included studies. While more evidence from prospective cohorts is needed, future studies should use novel statistical approaches to evaluate the relative contribution of Mn on MetS risk along with other inter-related exposures of nutrients or metals.

## Figures and Tables

**Figure 1 nutrients-14-00825-f001:**
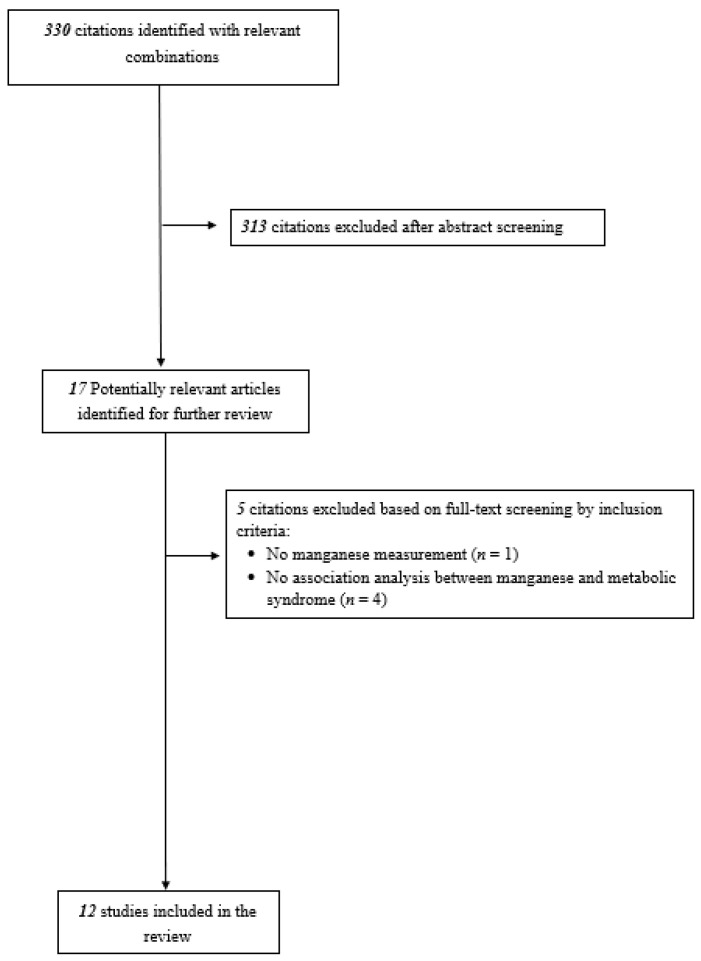
Flow of study selection.

**Figure 2 nutrients-14-00825-f002:**
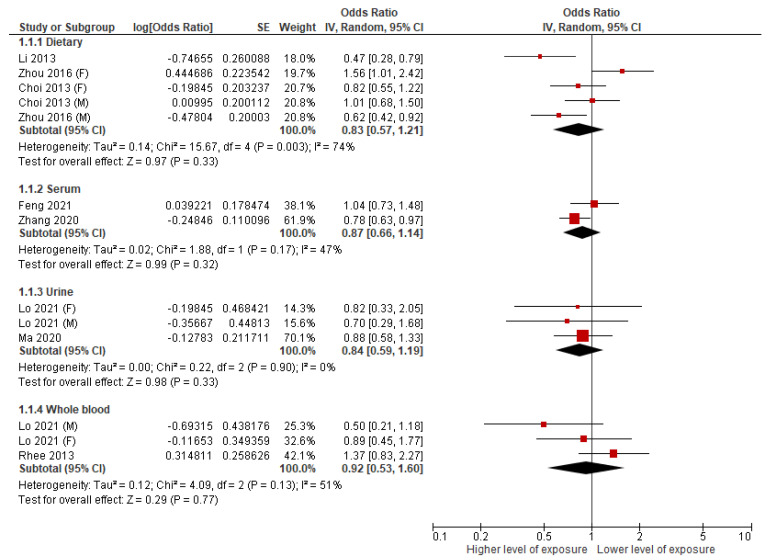
Forest plot for different exposures of manganese and metabolic syndrome. The Figure summarizes the overall association between the highest Mn level from diet, serum, urine, whole blood, and the likelihood of metabolic syndrome. Random effects models using the inversed variance (IV) approach were used to pool the estimates from individual studies. The effect estimates are presented as odds ratio with 95% confidence intervals (CI).

**Table 1 nutrients-14-00825-t001:** Description of the included studies.

Authors, Year	Country	Study Name	Study Design	Sample Size	Mean Age	Type of Mn Exposure	Definition of MetS	% of Male	Quality Assessment Scores ^a^
Bulka 2019 [20]	U.S.	U.S. NHANES 2011–2014	CS	1088	≥20	Whole blood	2009 Joint Scientific Statement of IDF, AHA/NHLBI, WFH, IASO	52.7	8
Choi 2013 [21]	Korea	The Korea NHANES 2007–2008	CS	5136	≥19	Diet	NCEP ATP III	40.6	8
Feng 2021 [19]	China	FAMHES	CS	1970	37.53	Serum	AHA/NHLBI	100	8
Ghaedrahmat 2021 [22]	Iran	Hoveyzeh cohort study	Nested CC	150	36–70	Urine	AHA/NHLBI	35.0	6
Li 2013 [23]	China	Nil	CC	544	53.7	Diet	NCEP ATP III	38.4	6
Lo 2021 [24]	U.S.	U.S. NHANES 2011–2016	CS	3335	≥18	Whole blood, Urine	NCEP ATP III	48.1	8
Ma 2020 [25]	China	Wuhan–Zhuhai cohort	CS	3272	53.2	Urine	NCEP ATP III	31.5	8
Rhee 2013 [26]	Korea	The Korea NHANES 2008	CS	1405	≥20	Whole blood	NCEP ATP III	49.3	8
Wen 2020 [27]	China (Taiwan)	Nil	CS	2444	55.1	Urine	NCEP ATP III	39.9	7
Zhang 2020 [28]	China	Beijing Population Health Cohort study	Nested CC	4134	60.0	Serum	IDF	49.5	6
Zhou 2016 [29]	China	CNNHS 2010–2012	CS	2111	53.1	Diet	AHA/NHLBI	47.2	8
Zhou 2021 [30]	U.S.	U.S. NHANES 2011–2018	CS	23,825	≥18	Whole blood	IDF	48.4	8

AHA/NHLBI: The American Heart Association and the National Heart, Lung, and Blood Institute; FAMHES: Fang Chenggang Area Male Health Examination Survey; IASO: International Association for the Study of Obesity; IDF: International Diabetes Federation; CC: case-control; CNNHS: Chinese National Nutrition and Health Survey; CS: cross-sectional; MetS: metabolic syndrome; Mn: manganese; NCEP ATP III: The National Cholesterol Education Program Adult Treatment Panel III; NHANES: National Health and Nutrition Examination Survey; U.S.: United States; WHF: World Heart Federation ^a^ Quality assessed by The Newcastle–Ottawa Scale (maximum score = 10 for cross-sectional studies and 9 for case-control studies).

## Data Availability

Not applicable.

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
