# Peer review of "Manganese Exposure and Metabolic Syndrome: A Systematic Review and Meta-Analysis"

_nutrients, 2022, doi:10.3390/nu14040825_

Round 1
Reviewer 1 Report
The review "Manganese exposure and metabolic syndrome: a systematic re-2 view and meta-analysis", by Wong and colleagues is concisely done and has value to the scientific community. However, the conclusion was that there is no association. Instead, they found higher dietary Mn may associate with a lower risk of MetS. This needs to be explained critically, as this might encourage Mn supplementation.
I have the following minor concerns:
- the descriptions of the original research on the four respective paragraphs (3.2 - 3.5) are too short, particularly for the meta-analysis section.
- The Figure legends have no text, only titles. Please add comprehensive legends that are intelligible.
- Although the review focussed on metabolic syndrome, a line or two on Parkinson's disease and Mn toxicity would help (because it was mentioned in the introduction).
Author Response
Reviewer 1
The review "Manganese exposure and metabolic syndrome: a systematic review and meta-analysis", by Wong and colleagues is concisely done and has value to the scientific community. However, the conclusion was that there is no association. Instead, they found higher dietary Mn may associate with a lower risk of MetS. This needs to be explained critically, as this might encourage Mn supplementation.
Response: Thank you for the comment. We only found inverse association between dietary Mn and MetS in some of the included studies, therefore we have clarified this observation in abstract and conclusion.
I have the following minor concerns:
- the descriptions of the original research on the four respective paragraphs (3.2 - 3.5) are too short, particularly for the meta-analysis section.
Response: Thank you for the comment, apart from presenting the results from each included study, we have now elaborated on the heterogeneity and consistency of the meta-analysis.
- The Figure legends have no text, only titles. Please add comprehensive legends that are intelligible.
Response: Thank you for the suggestion. Figure 1 is the flow of study selection and is self-explanatory, therefore we have added figure legend for Figure 2.
- Although the review focussed on metabolic syndrome, a line or two on Parkinson's disease and Mn toxicity would help (because it was mentioned in the introduction).
Response: Thank you for the suggestion, in the second paragraph of introduction, we have added several sentences to highlight the potential linkage between Mn toxicity and PD.
Reviewer 2 Report
This manuscript describes a meta-analysis on the association of manganese (dietary, serum, blood or urine) with metabolic syndrome. This is the first meta-analysis on this subject to my knowledge. The authors detected no significant association of manganese with metabolic syndrome. The methodology is sound, the results are clearly presented, and the manuscript is well-structured and succinct. The discussion is balanced well with the results of the meta-analysis. English language is decent but requires some work.
I have no major critiques but the following typos etc. were detected.
Lines 17-19: The highest Mn level from… ….were insignificant (should be “The association of the highest Mn level …. were insignificant”)
Line 31: NHANES is missing the word Nutrition
Line 43 : lasts for -> is
Table 1: CS and CC are not explained
Table 1: Wen et al was performed in Taiwan, so a different jurisdiction to mainland China with different laws and regulations which might affect the results. Should be differentiated, e.g. China (Taiwan) or China (Taipei) or Chinese Taipei.
Line 206: accelerates -> accelerate
Line 207: increases -> increase
Line 210: increases -> increase
Line 224: metals and neurotoxins -> metal and neurotoxin
Author Response
Reviewer 2
This manuscript describes a meta-analysis on the association of manganese (dietary, serum, blood or urine) with metabolic syndrome. This is the first meta-analysis on this subject to my knowledge. The authors detected no significant association of manganese with metabolic syndrome. The methodology is sound, the results are clearly presented, and the manuscript is well-structured and succinct. The discussion is balanced well with the results of the meta-analysis. English language is decent but requires some work. I have no major critiques but the following typos etc. were detected.
Response: Thank you for the compliment.
Lines 17-19: The highest Mn level from… ….were insignificant (should be “The association of the highest Mn level …. were insignificant”)
Response: We have revised accordingly.
Line 31: NHANES is missing the word Nutrition
Response: We have revised accordingly.
Line 43 : lasts for -> is
Response: We have revised accordingly.
Table 1: CS and CC are not explained
Response: We have indicated in the footnote that CS refers to cross-sectional, CC refers to case control.
Table 1: Wen et al was performed in Taiwan, so a different jurisdiction to mainland China with different laws and regulations which might affect the results. Should be differentiated, e.g. China (Taiwan) or China (Taipei) or Chinese Taipei.
Response: We have indicated China (Taiwan) as advised.
Line 206: accelerates -> accelerate
Response: We have revised accordingly.
Line 207: increases -> increase
Response: We have revised accordingly.
Line 210: increases -> increase
Response: We have revised accordingly.
Line 224: metals and neurotoxins -> metal and neurotoxin
Response: We have revised accordingly.